# Design of Industrial Standards for the Calibration of Optical Microscopes

**DOI:** 10.3390/ma14010029

**Published:** 2020-12-23

**Authors:** Alberto Mínguez Martínez, Cecilia Gómez Pérez, David Canteli Pérez-Caballero, Laura Carcedo Cerezo, Jesús de Vicente y Oliva

**Affiliations:** 1Laboratory of Metrology and Metrotechnics (LMM), School of Industrial Engineering, Technical University of Madrid, José Gutiérrez Abascal, 2, 28006 Madrid, Spain; cecilia.gperez@alumnos.upm.es; 2Laser Center, Technical University of Madrid, Campus Sur, Edificio “La Arboleda”, Alan Turing, 1, 28031 Madrid, Spain; David.canteli@upm.es; 3Spanish Metrology Center (CEM), Alfar, 2, Tres Cantos, 28760 Madrid, Spain; lcarcedo@cem.es

**Keywords:** metrology, traceability, calibration, microscopy, surface texture, standard

## Abstract

One of the most important fields of study in material science is surface characterization. This topic is currently a field of growing interest as many functional properties depend on the surface texture. In this paper the authors, after a short a review of different methods for surface topography characterization and the determination of the traceability problems that arise in this type of measurements, propose four different designs of material standards that can be used to calibrate the most common optical measuring instruments used for these tasks, such as measuring microscopes, metallurgical microscopes, confocal microscopes, focus variation microscopes, etc. The authors consider that the use of this type of standards (or others similar to them) could provide a step forward in assuring metrological traceability for different metrological characteristics that enables a more precise measurement of surface features with optical measuring instruments. In addition, authors expect that this work could lay the groundwork for the development of custom standards with specialized features tuned to gain a better metrological control when measuring specific geometrical surface properties.

## 1. Introduction

Nowadays material characterization is the foundation of science and industrial development. In this field, the different fields of metrology, the science of measurement, play an important role in helping researchers and engineers study and work on:Understanding the main features of materials such as their surface geometry, internal structure and chemical composition.The design and development of new materials, applications and technologies using all the knowledge obtained from the study of materials.

In addition, the current tendency in industry is towards the miniaturization of systems and materials and the fabrication of parts at the micro- and nano-scale. For this reason, metrology is a powerful tool to ensure more stable production, to reduce scrap and the cost of non-conformities. By controlling them, it is possible to increase the knowledge about materials science and industrial processes and to carry out continuous improvements. One particular field of interest is the study of surface texture, where interest in its characterization and control have increased dramatically recently [1,2,3]. A correct surface control can define the functionality of the product and affects performance and life service. Surface measurements can also improve the productivity and increase product quality. Nowadays, this characteristic is even more important due to the evolution of industry and the development of Industry 4.0 where accurate measurement became capital, as it is based on digital models of production and processes [4]. The part of the metrology that covers this field is dimensional metrology (DM) and its branch coordinate metrology (CM). This branch of metrology provides the traceability of length, area, and volume measurements, referring them to the unit of length (the meter) of the International System of Units (SI). It is used in many industrial applications to assure the quality and fit of products [1,5,6,7].

Most manufactured parts depend on the control of their surface characteristics [1,8]. At this point, it is important to define the concept of *surface topography*. It is understood as the overall surface structure of a part, the shape of a part and the feature that remains after manufacturing processes, also known as *surface texture* [8,9]. It is a key factor that affects the material’s behavior and many properties like tribology, optical properties and biocompatibility, among others [3]. Surface characterization is crucial for the comprehension of the geometry of the feature on a material, component, or device, as it is the interface that interacts with the environment and other components. In a composite material, the importance of studying surface texture (i.e., the size and angle of a layer, the position of fibers, the size of a pore, etc.) could allow us to understand their global properties [5].

The aim of this paper is:To rise the problem of traceability for surface texture measurement, especially when working with pieces at micro- and nano-scale, and the difficulty to accomplish the characterization of them.To present the different standards that can be found, the main materials of which they are made and the classic manufacturing processes.Taking into account the tendency towards miniaturization, the authors consider that in the near future the characterization of surfaces and materials will need specialized standards with custom shapes. For this reason, in this paper the authors propose different initial designs of custom standards to carry out measurements of some relevant metrological characteristics, such as distances, radii, angles, curvatures and customized shapes. These standards pretend to be the first step on developing customized standards for specialized applications.

### 1.1. Surface Topography Characterization

As previously seen, it becomes especially important to carry out an analysis of the surface for the characterization of new materials. The task of DM is measuring distances, angles and other dimensions, whose unit is the meter, and surface texture characterization. There are a great number of surface topography measuring instruments. According to the literature, measuring instruments can be divided into three classes, depending on the type of measurement method [1,10]:*Line-profiling method*: produces a two-dimensional graph or profile of the surface topography as a measurement result, representing the data as a height function z(x) vs. horizontal displacement x.*Areal-topography method*: produces a three-dimensional plot of the surface topography as a measurement result, representing the data as a height function of two independent variables z(x,y).*Area-integrating method*: Unlike the previous two methods where the measurement results was a mathematical function, z(x) or z(x,y), area integrating methods produce scalar results, which depend on the properties of the surface texture, from a representative area of a surface.

For each method, there are several techniques and measuring instruments to carry out the measurements. It is important to note that the areal-integrating method do not produce line profile data nor areal topography data. For this reason, it has been necessary to use those methods with calibrated roughness comparison specimens or other measuring instruments to distinguish the surface texture [10]. Otherwise, areal measurements give a more realistic interpretation of the whole surface and are more statistically significant [1]. These methods, collected in Figure 1, could be also classified into contact methods and optical methods [2,11].

There is a capital problem with profiling methods because the use of contact methods can modify and also destroy structures and materials at the micro- and nano-scale. Due to this, it becomes necessary to use non-contact, i.e., non-destructive, exploration techniques and here is where optical measuring systems show their full potential. Another advantage of optical methods is that the scanning is faster than in contact methods. Conversely, these methods are sensitive to several surface qualities and surface slope that could affect to the accuracy of the measurement [11]. Considering the incoming importance of micro- and nano-scale systems and technologies, the relevance of optical measuring methods is even greater. For this reason, the development of metrology for optical measuring instruments is one of the more active working fields in DM. Optical microscopes are specialized instruments that permit obtaining magnified visual or photographic images of surfaces and small objects [12,13]. These devices have been extensively used in several research fields, such as biology, medicine, mineral, metallurgy, machinery and others [14]. One particular case are metallographic microscopes. This kind of microscope uses the light that is reflected by the sample to obtain qualitative characteristics of metals and alloys, and they have been traditionally used for material characterization [15].

### 1.2. Traceability and Calibration of Optical Methods

The calibration and traceability for surface texture measuring instruments have been a subject of research during the last century and is currently an important topic, because the possibility to “visualize” the surfaces at micro- and nano-scale is going to be very important for the development of manufacturing processes and quality control in the near future. For a better understanding, it is important to define these metrological concepts. In the metrology field, definitions are collected at the International Vocabulary of Metrology (VIM) [16]:Metrological traceability is defined at Section 2.4.1 as “*the property of a measurement result whereby the result can be related to a reference through a documented unbroken chain of calibrations, each contributing to the measurement uncertainty*”. The sequence of measurement standards and calibrations that enables to relate the result of a measurement to a reference is known as metrological traceability chain (Section 2.4.2, [16]).Calibration is defined at Section 2.3.9 as “*the operation that, under specified conditions, in a first step, establishes a relation between the quantity values with measurement uncertainties provided by measurement standards and corresponding indications with associated measurement uncertainties*”.

There are many problems that remain unsolved and it is necessary to develop new procedures and methods for the calibration of optical measuring instruments [1,17]. For this reason, there are several authors and research groups focused on this topic and the normative that covers this subject is continuously under development and revision. One of the main problems is the development of material standards to transmit traceability to optical measuring instruments. In this context, the concept of surface standards becomes of interest.

## 2. Commercial Surface Standards

A measurement standard is defined at Section 5.1. of [16] as “*the realization of the definition of a given quantity, with stated quantity value and associated measurement uncertainty, used as a reference*”. During the twentieth century, the development of material standards and instruments in the field of surface metrology (roughness and form measurements) was led by the Taylor-Hobson company (Leicester, U.K.), which was the principal manufacturer of stylus instruments. In the 1940s, they started to produce a roughness standard for their calibration, which was considered as the first practical standard. This specimen was a regular profile roughness with flat-bottomed grooves, and this type of standard is defined in ISO 5436 as a type B. In 1965, Häsing and his team developed random profile roughness specimens with Ra range from 0.15 to 1.5 µm at the Physikalisch-Technische Bundesanstalt (PTB, the German national metrology institute) (Braunschweig, Germany). Since then, the specimens developed at PTB have been widely used in the European calibration network and other researchers, like Song and Krüger, have designed other profiles that are currently gathered under Type D of ISO 5436 [18,19]. Another important national metrology institute, the National Physical Laboratory (NPL) (Teddington, UK) in the UK produces certain types of standard also focused on the traceability and the maintenance of the traceability chain [2,20]. Typically, these standards consist of a base and a hardened layer where the exploration surface is marked as shown in Figure 2, but it is also possible to find standards made of solid material [21,22].

In this section, the authors present the most common materials used for surface standards, the manufacturing processes used to produce them and the shapes of commercial ones. It is important to note that it is difficult to find enough information about these topics in literature. All the information presented at this point has been taken from normative, technical books and suppliers’ web pages.

### 2.1. Materials

The first step of the manufacturing process is the selection of the correct material and it could be a difficult step. According to the International Organization of Legal Metrology (OIML), it is necessary to pay attention to the requirements listed below for the fabrication of dimensional metrology standards [23]:The chosen materials should, under normal conditions of use, be durable, stable and resistant enough to environmental influences.The expansion related to temperature changes of ±8 °C from the reference temperature should not exceed the maximum permissible error for the accuracy class to which the etalon belongs.If the measurement is carried out under a specified tension, variations of ±10% should not produce a variation in length exceeding the maximum permissible error.

According to this, there are a lot of materials and possibilities. It is important to note that in profiling measurement processes, the standard will be run through a stylus that can scratch it, so that it will be very important to have a material with high hardness. However, as commented above, the tendency of industry is to miniaturization of products and materials and, therefore, optical measurements are very important to preserve the surface structure [1]. In addition, these systems are fast and have high resolution, qualities that it seems that will be very important soon. In this case, the hardness of the material is less relevant, so that it is possible to use other materials. It is also possible to reach a compromise solution that enables to calibrate the metrological characteristics of areal surface topography measuring instruments with optical and contact measuring instruments, as in [24,25], commented below. Table 1 collects the most common materials used for roughness standards according to the information obtained from several suppliers, technical books and articles and MATWEB, which is a very useful database for material engineers [22,23,25,26,27,28,29,30,31,32,33,34,35,36]:

### 2.2. Manufacturing Processes

Once the material is selected and prepared, it is time to engrave the desired shape on the standard. Here is even less information in literature about the marking process. After consulting some specialists, belonging to different National Measuring Institutes (NMIs) and calibration laboratories, it seems that there are two main processes to mark the profile in the specimen:*Pressure method*: this method consists on heating the material and, then, a mold with the shape (the negative) is put over the surface and, by applying pressure, the profile is marked over the surface.*Grinding method*: this method consists on griding the surface with the negative in the same direction, make a roughness test and repeat once and again until the expected roughness is achieved.*New technologies*: there are several modern technologies such as silicon micro-technologies, lithography, focused ion beam, direct laser writing and other additive manufacturing (AM) technologies that may help metrology industry on manufacturing new standards.

### 2.3. Traditional Shapes for Surface Standards

Traditionally, and according to ISO 5436-1, five different types of artifacts have been used for the calibration of profiling measuring instruments [18]:*Type A*: Depth measurement standard for calibrating the component on *z*-axis (vertical). They could be *wide grooves with flat bottoms* (Type A1) or *wide grooves with rounded bottoms* (Type A2)*Type B*: Tip condition measurement standard for calibrating the condition of stylus tip. They could be standards with one narrow groove with rounded bottoms or a sequence of separated grooves with increasing dimensions (Type B1), standards with two groove patterns with the same Ra value but one sensitive and other insensitive to the dimensions of the tip (Type B2) or standard with a fine protruding edge (Type B3).*Type C*: Spacing measurement standard for calibrating vertical and horizontal profile components. They are composed by a grid of repetitive sine wave profile (Type C1), isosceles triangular profile (Type C2) simulated sine wave grooves (Type C3) or arcuate grooves (Type C4).*Type D*: Unidirectional irregular profile for overall calibration of profile measuring instruments. They could be standards with unidirectional irregular profile that simulates workpieces (Type D1) or standards with circular irregular profile with a near-constant cross-section along the circumference (Type D2).*Type E*: Profile coordinate measurement standard for calibrating the profile coordinate system of the measuring instrument. They could be standards that consist of a sphere or hemisphere, called precision sphere or hemisphere (Type E1) or precision prism with a trapezium cross-section (Type E2). The schemes of these standards are shown in Table 2 [8,18,37].

ISO 12,179 [37] describes how to use these standards for calibrating profile measuring instruments. However, it is possible to adapt the calibration procedures to calibrate optical measuring instruments with them by operating in profile mode. As an example, one optical method of surface texture measurement for confocal microscopes has been recently developed. In this procedure, the plane XY and *Z*-axis and the Ra (roughness parameter that it is given by the confocal microscope) are traced to the SI reference. This method, that could be adapted for other types of microscopes with minor changes, is based in the use of traceable standards and parts easy to find at industry and roughness is calibrated using Type C1 and D1 standards [38]. 

It is also possible to find suppliers that provide standards with special shapes corresponding to the result of machining processes, as shown in Figure 3. Typically, this kind of standards are used to easily estimate the roughness of a machined surface. Six machining methods are mainly considered: turning, vertical milling, horizontal milling, grinding, flat lapping and reaming [39].

For areal surface texture measurements and calibrations, ISO 25178-701 describes six different artifacts [1,40]:*Type ER1*: Measurement standards with two parallel grooves. This kind of standard is used for calibrating the vertical and the horizontal amplification coefficients of the measuring instrument.*Type ER2*: Measurement standards with four grooves forming a rectangle. This kind of standards is used for calibrating the vertical and the horizontal amplification coefficients and the perpendicularity of x and y-axes of the measuring instrument.*Type ER3*: Measurement standards with circular groove. This kind of standards is used for calibrating the vertical and the horizontal amplification coefficients and the perpendicularity of x and y-axes of the measuring instrument.*Type ES:* Measurement standards with a sphere/plane intersection. This kind of standards is used for calibrating the vertical and the horizontal amplification coefficients, the response curve of the probing system, the perpendicularity of x and y-axes of the measuring instrument, the geometry of the stylus, the radius of the stylus tip and the cone angle.*Type CS*: Measurement standards with a contour profile. This kind of standards is used for overall calibration along one lateral axis of the measuring system.*Type CG1*: X/Y-Crossed-grating measurement standard. This kind of standards is used for calibrating the horizontal amplification coefficients and the angle between x and y-axes of the measuring instrument.*Type CG2*: X/Y/Z-Crossed-grating measurement standard. This kind of standards is used for calibrating the vertical and the horizontal amplification coefficients and the angle between x and y-axes of the measuring instrument.

The schemes of these standards are shown in Table 3 [1,40]. As seen previously, an “*Areal calibration set*” was developed at NPL to calibrate surface topography measuring instruments. This measuring instrument enables users to [24]:Calibrate common optical instruments equipped with 10×, 20× and 50× lenses and stylus instruments.Meet the requirements of ISO standards for determination of three-dimensional surface texture.Validate instrument performance.Adopt good measurement practices and automate calibration procedures.

As it is possible to see in Figure 4, this standard is characterized by its five types of calibrated patterns [24]:*Type PGR*: this part of the calibration set has the feature of a step height standard with a rectangular groove. There are six depths: 50 nm, 100 nm, 200 nm, 500 nm, 1 µm and 2 µm. This type of pattern allows to address the amplification coefficients ($a_{x}$,$\;a_{y}\;$,$\;a_{z}$), the linearity ($l_{x}$,$\;l_{y}\;$,$\;l_{z}$) and the perpendicularity of X and Y-axes ($\Delta_{PER}{}_{xy}$) in conjunction with Type ACG patterns.*Type ACG:* this part of the calibration set consists of a cross grating pattern with three different pitches: 20 µm, 50 µm and 100 µm. This type of pattern allows to address the amplification coefficients ($a_{x}$,$\;a_{y}\;$,$\;a_{z}$), the linearity ($l_{x}$,$\;l_{y}\;$,$\;l_{z}$) and the perpendicularity of X and Y-axes ($\Delta_{PER}{}_{xy}$) in conjunction with Type PGR patterns.*Type AFL*: this part of the calibration set consists of a flat surface for flatness measurement. The parameter $S_{z}$ is lower than 20 nm. This type of pattern allows to address the flatness deviation ($Z_{FLT}$) and the measurement noise ($N_{M}$).*Type ASG*: this part of the calibration set is for controlling the resolution thanks to its star shape grooves. There are four different dimensions: from 0.7 µm to 50 µm with depth of 200 nm, from 0.7 µm to 25 µm with depth of 200 nm, from 0.7 µm to 25 µm with depth of 50 nm and from 0.2 µm to 6 µm with depth of 50 nm. This type of pattern allows to address the lateral period limit ($D_{LIM}$).*Type AIR*: this part of the calibration set is an irregular area to enable areal parameter measurements (Sa, Sq, Sz, Ssk and Sku) with 40 or 70 µm of autocorrelation length.

Another different, remarkable surface standard is made by SiMETRICS GmbH (Limbach-Oberfrohna, Germany) for Sa parameter determination. This parameter is the equivalent of the roughness parameter Ra for surface texture. As shown in Figure 5, they are called Roughness Standard ARS (areal) and Type ADT [26]:

The roughness standards have different kind of markers [26]: *Type ARS or Type c (coarse)*: Used when the roughness parameter Sa is higher 100 nm. The pattern is a quadratic field with an edge length of 5.6 mm.*Type ADT or Type f (fine)*: Used when the roughness parameter Sa is lower 100 nm. This pattern has 16 quadratic fields with an edge length of 1.75 mm. In the red lines of the 4 fields along the diagonal used for the measurement with interferometric or confocal microscopes or an AFM.

Recently, a new model of standard for measuring instrument characterization has been developed at PTB. It is called *chirp standard* [27]:

This model of standard is applicable for characterizing the Instrument Transfer Function (ITF) of areal surface topography measuring instruments as well as its angular-dependent asymmetries. The chirp standard presented in Figure 6 has two main innovations:
The circular structure patterns, preferred for characterizing the ITF features in different angular directions to detect angular-dependent asymmetriesThe feature scheme is generated by rotation of a horizontal linear chirp profile around the *z*-axis.

Additionally, this standard presents a surface with a predefined band spectrum ($\lambda_{min}$ … $\lambda_{max}$) with a quasi-flat amplitude. The material composition of the top and bottom surfaces of the features are the same, avoiding the influence of the phase jump issue of optical reflection. The prototype was made of Niobium on silicon, with patterns in a spectral range $\lambda_{\min}\;=\;1.0\;\mu m$ and $\lambda_{\max}\;=\;16.0\;\mu m$ and was manufactured using electron-beam lithography combined with reactive ion etching technique [27].

Also at PTB’s standard list, it is possible to find a Cu depth-setting standard, in Figure 7. The main advantage of this model is that it is formed by grooves with the same slope of the walls but having different depths [23]. 

The main characteristics of this standard are:It is made of nickel on copper.It is formed by grooves with 55° slope µm and a flat bottom.The grooves are separated 400 µm.

## 3. Design of Customized Surface Standards

As stated above, the tendency to miniaturization makes necessary to adapt and design measuring instruments and procedures. In this section authors propose the characteristics of four designs of standards for dimensional measurements and material characterization in optical microscopes. Each design includes the main characteristics of the standards, calibration strategies and some applications to facilitate future work.

The principles followed for the designs were:As CM is extensively developed up to millimeter scale, the designs should adapt its fundamentals to micro- and nano-scale.The designs should be based on the standards presented above.The designs are conceived only to give traceability to metrological characteristics on XY plane. The traceability for *Z*-axis should be accomplished by using step height standards or similar.The standards should be traced by using classical measuring instruments, such as stylus instruments and profile projectors.These standards are designed to be used for image analysis.The proposed calibration is designed considering inversion techniques.

Four different standard designs have been developed and their main characteristics will be described in this section. Note that the designs could be adapted for different applications.

### 3.1. The Grid

In this design a region of the surface is divided in 100 different sectors, forming a square cross-grating with 10 sections for side. The model is based on standard type CG1 of [40] but can be reconfigured depending of the surface that is going to be explored. The main objective is to carry out the characterization of each sector having the position controlled.

In Figure 8:*a*: is the size of each square, i.e., the separation between two consecutive lines.*b*: is the length of the lines, that is *12·a*.*c*: is the length of the central lines, that is *15·a*.

In this model, there are two perpendicular arrows at the lower, left corner. This allows the operator to zero in the same position. The purpose of *b* and *c* dimensions is to facilitate the control and calibration of the standard with contact methods, when necessary, without damage the exploration surface. 

The calibration of this model would be accomplished by carrying out the measurement of the pitch on one direction, turning the standard 45° four times and repeating the measurement of the pitch on each direction, as in [38]. Typically, several measurements should be taken for obtaining better results. Note that to characterize this design it will be necessary to pay attention to the following sources of error [40]:$a_{x}$: the horizontal pitch.$a_{y}$: the vertical pitch.θ: angle between X and Y-axes.The parallelism between lines.The width of the lines.The straightness of the lines.

When the calibration process is carried out using contact methods, it will depend on the dimensions and the shape of the stylus and it would be difficult to obtain good results of the measurement in some circumstances. For this reason, the use optical measuring instruments for the calibration as in [38] is a good alternative. 

When the standard is calibrated, it can be used to study the position and dimensions of features or defects on surfaces. In metallography, this design may help on grain size study. In composite materials analysis, this design may allow to check the distribution of fibers or layers. These two applications are capital to understand the material properties. If the intersection between lines is considered, this design can act as a ball pattern and can be used in a similar way as ball plates are used in CM Machines (CMMs). In micro- and nano-scale AM parts, this design would help to analyze the dimensions, angles and relative positions, adapting the principles of CM to this scale.

### 3.2. The Dartboard

This design divides the exploration surface in *n* different circular sectors, so that it would be possible to observe curves (circumferences) with the same diameter. The model presented in Figure 9 is based on standard types D2 of [18] and ER3 of [40]:

This model is very simple. It is characterized by the different diameters. The design includes a cross and a zero mark to facilitate the positioning during calibration. Note that to characterize this design it will be necessary to pay attention to the following sources of error:$a_{xi}$: the horizontal relative displacement of the centers.$a_{yi}$: the vertical relative displacement of the centers.Roundness errors.Errors in diameter.

For its use, it is necessary to calibrate each diameter. The calibration of this model would be carried out by using an optical measuring microscope, a vision measuring machine or even a profile projector in some cases if lines are graved on the surface a 3D stylus instrument could be used. Considering the case of profile projectors, the calibration process may consist of flushing the reference lines of the measuring instrument with each circle. First, the reference line is flush with the left side of the circle in the left to right direction and is defined as the zero. Then, moving the scanning table, the reference line is flush with the right side of the circle. The operation is repeated in the opposite direction. This data collection is repeated several times changing the orientation of the reference lines 45° three times (to 45°, 90° and 135°) for a good uncertainty estimation.

When the standard is calibrated, it can be used as a go no-go standard for round features using image analysis. In material analysis, using the suitable magnification it is possible to analyze the size of pores or similar characteristics. In micro- and nano-scale AM parts, this design would help to analyze diameters of holes and round parts.

### 3.3. The Star

This design is conceived to have angular references to characterize relative position of different features at exploration. The model presented in Figure 10 consists of a reference circumference, two perpendicular lines to remark vertical and horizontal directions, n lines corresponding to the different angles and a zero mark. The design would be supplied with a diagram where the angles in the standard should be included. 

Note that, when characterizing this design, it will be necessary to pay attention to the following sources of error:The angle of the lines.The width of the lines.The straightness of the lines.

The calibration of this model would be carried out by using an optical measuring microscope, a vision measuring machine, a profile projector or even a stylus instrument when the lines are engraved. In the case of profile projectors, the reference line is flush with zero and, then, said line is flush with each angular mark in an upward direction. When it is finished, it is carried out in a downward direction. This procedure is repeated several times for a good uncertainty estimation.

When the standard is calibrated, it can be used to study the relative position of surface features. In composite materials analysis, this design may allow to check the position of fibers or layers regarding a defined reference.

### 3.4. The Spiral

The design presented in Figure 11 consists of 90° circumference arches with increasing radius and two vertical lines and two horizontal lines:

The radius increases according to:(1)$$R_{n}=R_{0}+n\cdot l$$
where:$R_{n}$ is the radius of each arch.$R_{0}$ is the radius of the central circle.n is the number of times the spiral has crossed the horizontal or vertical lines.l is the proportional increase. It is defined by the side of the central square that defines the position of the different circle.

Note that to characterize this design it will be necessary to pay attention to the following sources of error:$a_{x}$: the horizontal error square side dimension.$a_{y}$: the vertical error square side dimension.Roundness errors.Errors in diameter.Errors in 90° angle definition.The dimension of the lines.The straightness of the lines.The angle between vertical and horizontal lines.

The calibration of this model would be carried out by using an optical measuring microscope, a vision measuring machine or even a profile projector. In the case of profile projectors, the calibration is accomplished flushing the reference line with the center and the intersection between the horizontal or vertical reference line and the circumference arch, as for dartboard design. When the standard is calibrated, it can be used for the study of curve features. 

## 4. Proof of Concept

In this section, the authors present an example of how the manufacturing process could be continued. As it is a proof of concept, it will not be calibrated but will be evaluated to be calibrated with a stylus instrument. The pattern will consist of four perpendicular lines two by two, separated by 50 µm. Once the design is selected, the next step is to determine the material from which it is made. For this task, it is necessary to consider the manufacturing and the calibration processes. As seen above, the standards are usually manufactured by engraving the desire shape over the exploration surface. However, using AM technologies, it would be possible to obtain the standard shape by growing it over a surface. Either by engraving the motif on the surface or constructing it by AM techniques, it is necessary to take also into account how it will be calibrated. For example, if a low uncertainty is required, the standard would be calibrated using a stylus instrument. This is because it is a well-known technology that represents the real surface. The problem is that, as it is a contact method, it may damage a standard manufactured by AM techniques. So that, in this case, it would be necessary to choose a material with high hardness where the standard will be engraved. 

### 4.1. Selection of the Material

Due to its hardness, its optical properties and the facility of acquiring it, authors selected polished monocrystalline silicon wafers as the material for the standard. In addition, as it is very brittle and could be difficult to handle without breaking it, it will be glued to a microscope slide. An scheme of the assembly is shown in Figure 12:

### 4.2. Selection of the Manufacturing Process

Considering the versatility of forms that can be obtained, authors propose to use laser techniques to make the grooves. Laser systems allow any type of curve to be drawn, no matter how complicated it may be. This flexibility would help manufacturers on obtaining custom surface standards that fit the requirements of the measuring process and the needs of traceability on surfaces. At this point, it is necessary to define the objective dimensions to determine the process parameters.

In microscopy field, stage micrometers have been traditionally used to give traceability to microscopes. This measuring instrument is a material measure of length. Typically, it consists of a scale where one millimeter is divided in a hundred parts. As it is widely used across the industry, authors consider this measuring standard will simplify this problem. Once a commercial metallic stage micrometer was analyzed at the stylus instrument, the graphic on Figure 13 was obtained. Note that it is represented the shape of the 101 lines (the colored curves) and the mean curve (the thick, black line) in this plot. The dimensions will be taken as reference for further work.

### 4.3. Results

After several tests, authors obtain the pattern on Figure 14. The pattern has been obtained using a HIPPO (Spectra Physics, Andover, MA, USA) that emits pulses of 15 ns at a wavelength of 355 nm. Note that the figures are taken with a DCM3D confocal microscope (Leica, Wetzlar, Germany) and analyzed with the SensoSCAN—LeicaSCAN DCM3D 3.41.0 software developed by Sensofar Tech. (Terrassa, Spain).

From the section shown in Figure 15, the software provides the following data:
The separation between the slopes is 5.97 µm.The depth of the groove is 1.03 µm.

As the stylus tip has a radius around 2 µm [41] it will be possible to calibrate it using a stylus instrument, so that it will be possible to obtain standards with low uncertainty.

## 5. Discussion

As a set of potential applications, authors would like to highlight the following:These custom standards can be used during calibration of optical measuring instruments to give traceability for dimensions on *XY*-plane, curvatures and angles.Selecting the suitable material, manufacturing process and pattern scheme for each application, it would be possible to create infinite types of standards with different properties and forms to be used micro- and nano-metrology.Defining a maximum and a minimum dimensions of a feature, it is possible to create go-no go standards that facilitate the measurement at industrial metrology laboratories. This application would be even more important because having go-no go standards that controls the acceptable dimensions make easier the monitorization of new manufacturing processes.With a correct design, these standards could be used at the study of the fracture on specimens after traditional destructive test, such as tensile, Charpy or torsion tests. The study of this geometry could be very important and interesting to understand the behavior of the materials under different efforts and, then, calculate and measure its properties on each direction.These standards can be used as a characterization tool of microstructures or surface features in materials, like peaks, valleys, and pores, when the surface is studied with a metallographic microscope. Correctly positioning the measurand and the standard in the workplace it is possible to carry out and study of the tilt of a surface, the thickness of a layer or even the pore size and distribution. It is possible to find freeware that eases the 2D analysis of microscope images [42,43,44,45].

The main problem that authors find is how to work with opaque materials. Since this kind of materials prevent the passage of the light through them, the calibration process and the operation with them will vary. Regarding the calibration process, the use of transparent materials allows measurements to be made by placing the standard turned over, obtaining a greater number of measurements and influencing the uncertainty. However, a standard made of an opaque material requires to modify the calibration process to obtain a comparable uncertainty. Regarding the operation with transparent standards, it would be possible to use them placing the standard between the objective and the sample. However, when opaque standards are used, it would be necessary to carry out an image analysis using a specialized software that allows to overlap both, the standard and the sample images. This situation could imply the necessity of calibrating the software and the development of complex measurement procedures for traceability insurance.

## 6. Summary and Conclusions

After evaluating the current needs of industry and the requirements that metrology will face in the near future, the authors consider that this article should be a good starting point for the design of custom standards for optical measuring instruments used for surface characterization. Although these designs are developed for measuring microscopes, they can be used in the calibration of measuring instruments with wider measuring fields. The authors also think that micro- and nano-scale materials and devices are on their first steps and metrology should go ahead of this development. In the next years it will be necessary to carry out measurements and characterization of materials with optical measuring instruments and the use of standards should be easy, agile, adaptive to different forms and, of course, traceable to the SI unit of length (the meter). In this paper, authors present four different models as a starting point for future work.

Additionally, a proof of concept pattern has been developed for validating the design process, choosing a material with good optical properties that enables the calibration with stylus instrument and selecting a technology that enables to make the grooves with the purposed dimensions and shapes.

## Figures and Tables

**Figure 1 materials-14-00029-f001:**
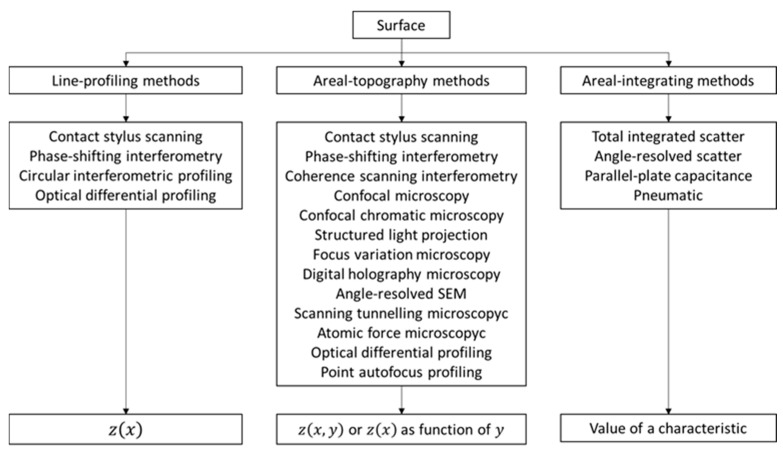
Surface characterization methods [10,11].

**Figure 2 materials-14-00029-f002:**
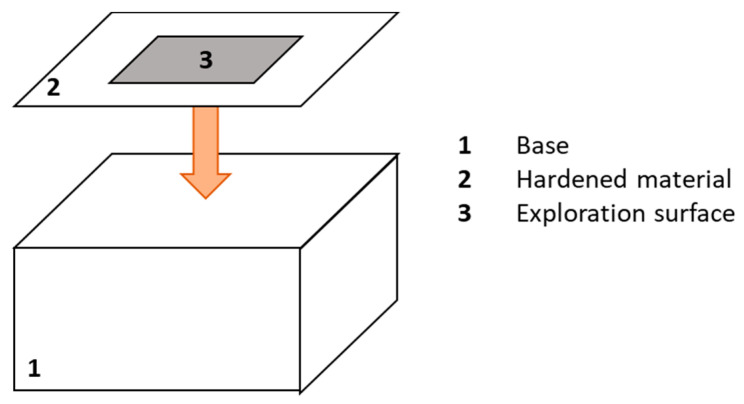
Roughness standard components [21,22].

**Figure 3 materials-14-00029-f003:**
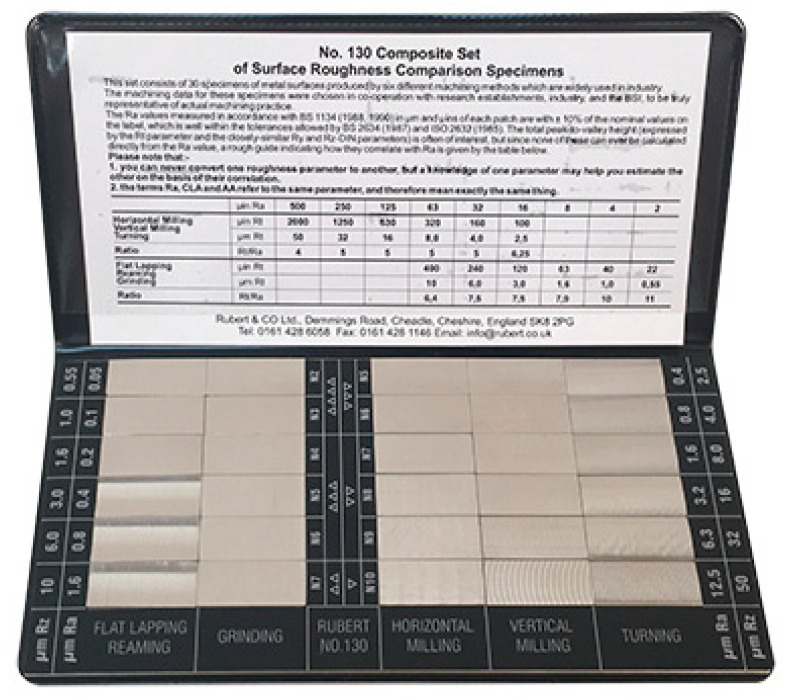
Commercial Roughness Comparison Specimens [39].

**Figure 4 materials-14-00029-f004:**
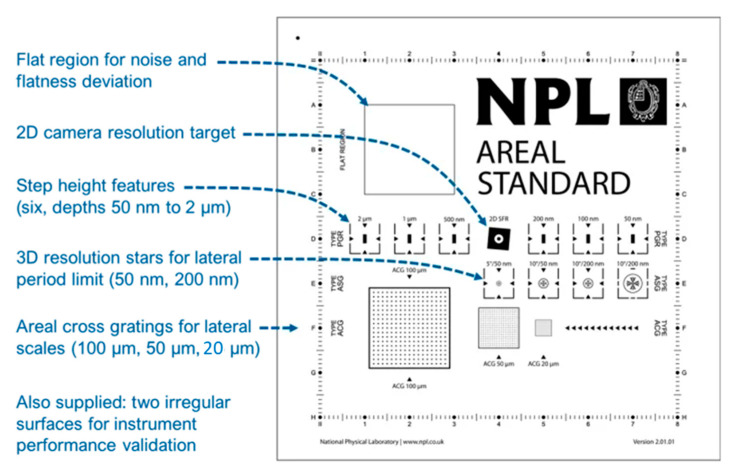
NPL areal standard [25].

**Figure 5 materials-14-00029-f005:**
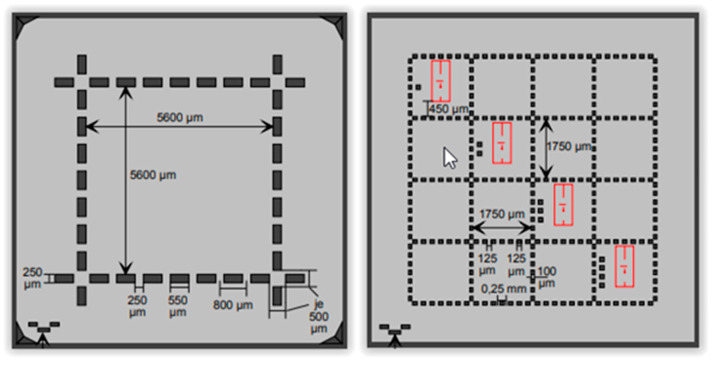
SiMETRICS Roughness Standard ARS (areal) Type ADT [26].

**Figure 6 materials-14-00029-f006:**
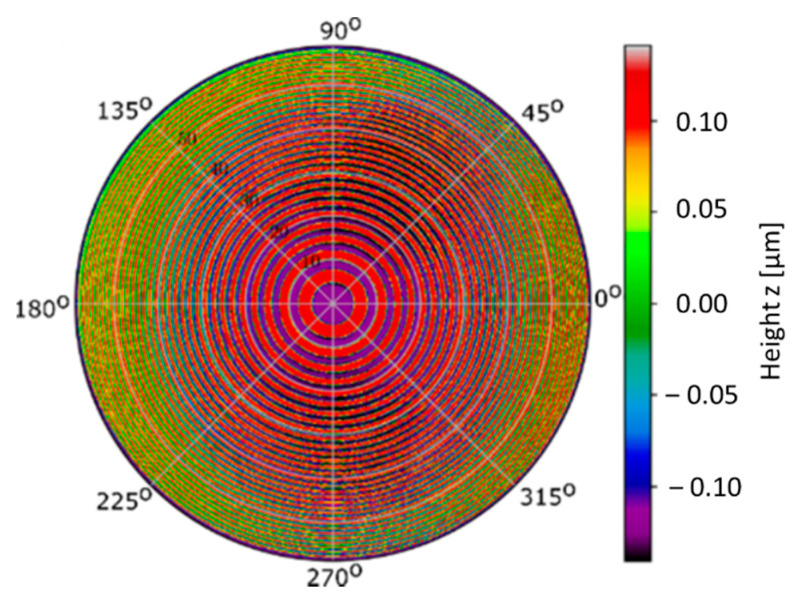
PTB chirp standard [27].

**Figure 7 materials-14-00029-f007:**
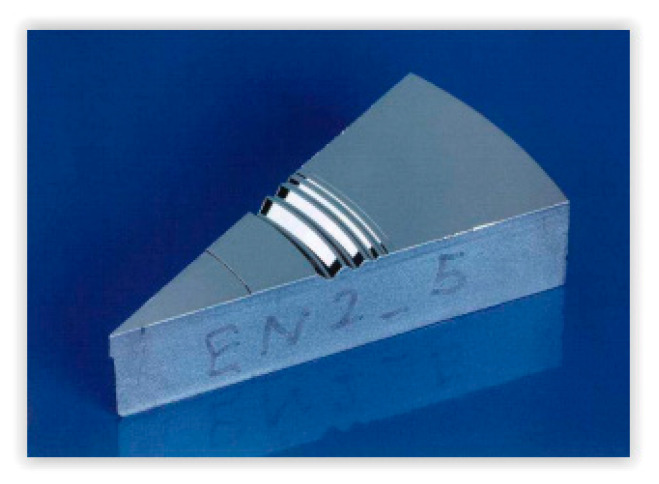
PTB Cu depth-setting standard [23].

**Figure 8 materials-14-00029-f008:**
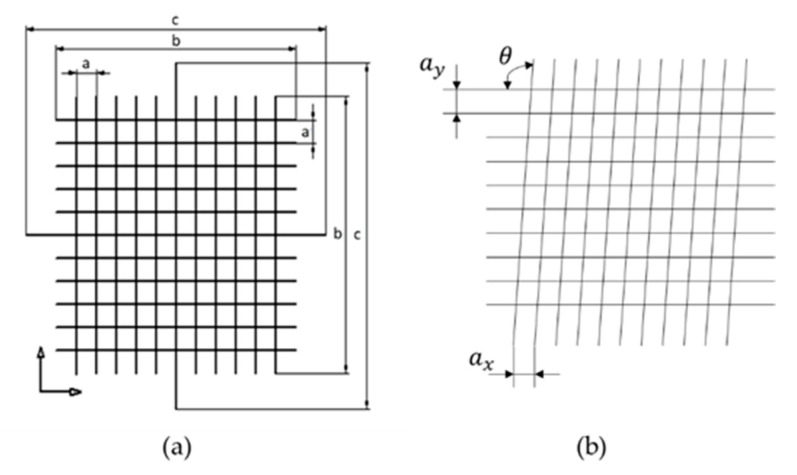
(**a**) The grid model and (**b**) it sources of error.

**Figure 9 materials-14-00029-f009:**
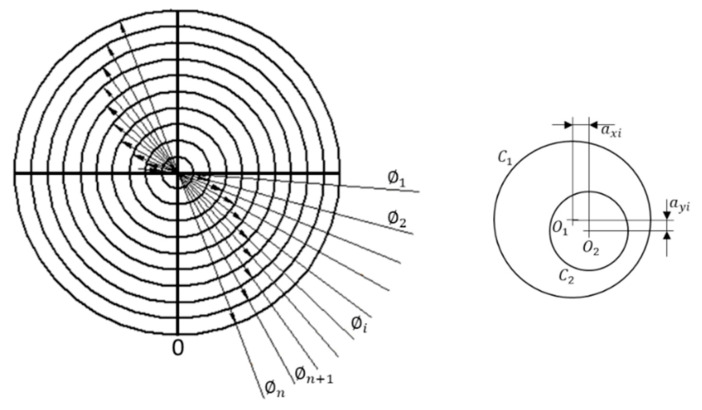
The dartboard model.

**Figure 10 materials-14-00029-f010:**
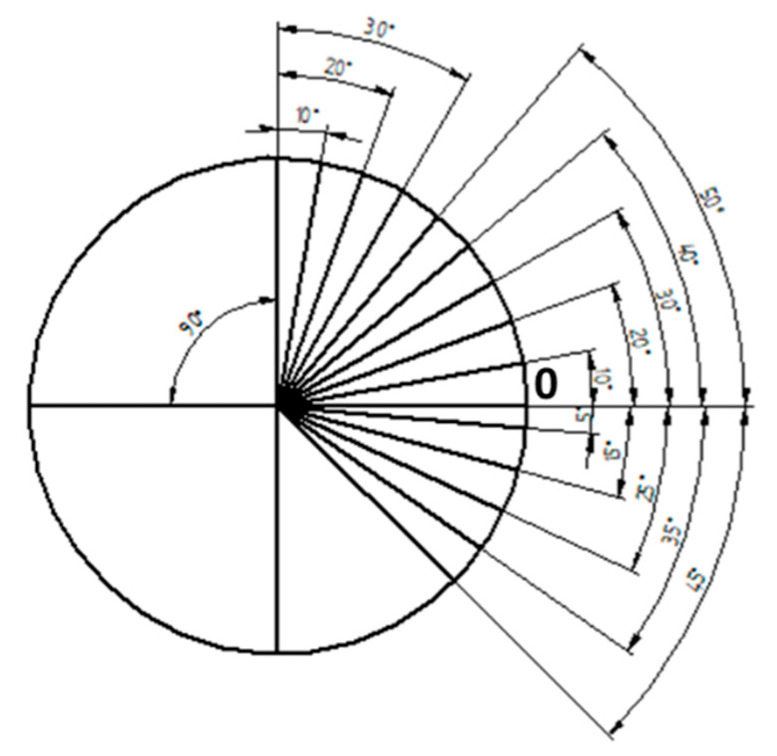
The star model (n=13 lines).

**Figure 11 materials-14-00029-f011:**
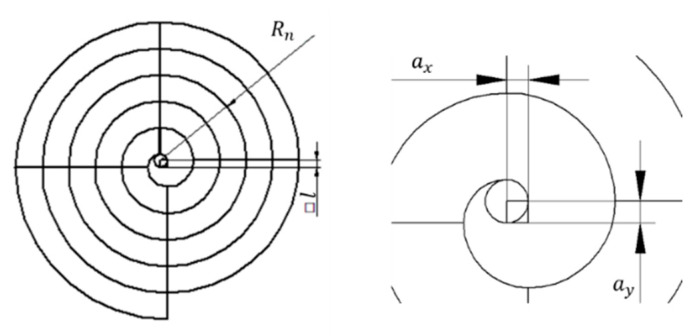
The spiral model.

**Figure 12 materials-14-00029-f012:**
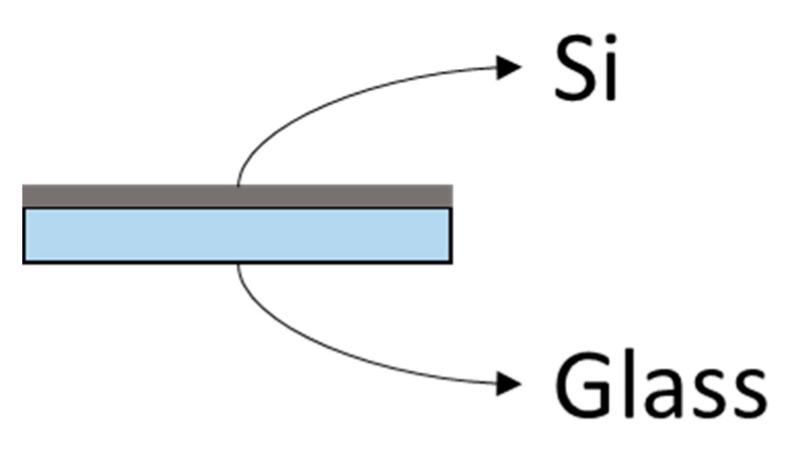
Materials for the proof of concept standard.

**Figure 13 materials-14-00029-f013:**
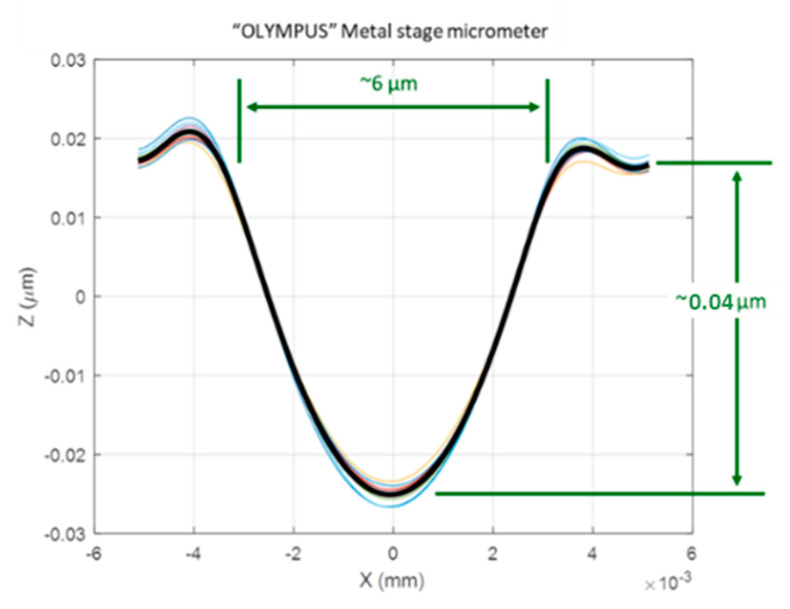
Grooves dimensions of commercial stage micrometer.

**Figure 14 materials-14-00029-f014:**
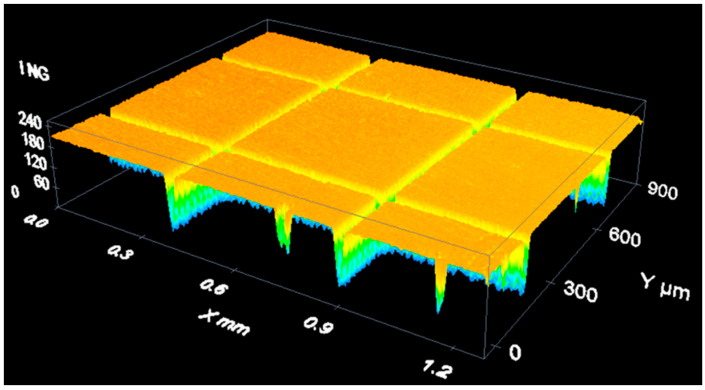
Proof of concept pattern seen in a confocal microscope at 10× magnification.

**Figure 15 materials-14-00029-f015:**
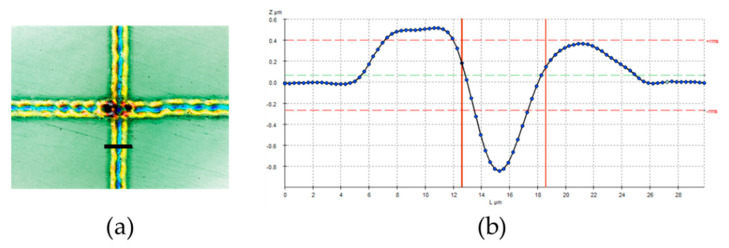
(**a**) Detail of proof of concept pattern seen in a confocal microscope at 50× magnification (**b**) Profile of the section.

**Table 1 materials-14-00029-t001:** Common materials for roughness standards.

Base	Exploration Surface	Hardness Vickers (HV)	Base	Exploration Surface	Hardness Vickers (HV)
Hardened stainless steel	Electroformed-nickel	540	Silicon	Niobium	130
Hardened stainless steel	Electroformed-nickel	540	Silicon dioxide	Chromium	1060
Hardened stainless steel	Hard-silver	220	Hard aluminum alloy	–	280
Hardened stainless steel	–	400	Mica	Colloidal Gold	216
Nickel	–	540	Glass	Silicon	850–1000^−2^
Nickel	Nickel-Boron	<1110 ^1^	Glass	Chromium	1060
Copper	Electroformed-nickel	540	Borosilicate glass	Lapped Silicon	850–1000^−2^
Copper	Copper	100	Polished glass	–	350
Silicon	–	850–1000 ^2^	Quartz	–	960
Silicon	Chromium	1060	Quartz	Silicon	850–1000^−2^
Silicon	Platinum	549	Quartz	Silicon die on Quartz	850–1000^−2^
Silicon	Iridium	1760	Quartz	Chromium	1060
Silicon	Tungsten	3430	Plastic	Polished glass	350

^1^ Conversion of [31] data with [32] for a better comprehension. ^2^ The hardness value of silicon has an extreme dependence on the manufacturing process used.

**Table 2 materials-14-00029-t002:** Surface texture artifacts according to ISO 5436-1 [8,18,37].

**Standard**	**Scheme**
Type A1	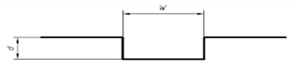
Type A2	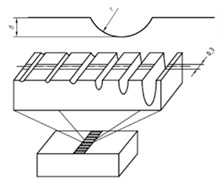
Type B2	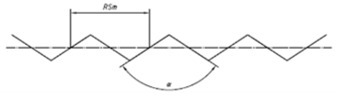
Type B3	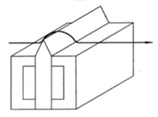
Type C1	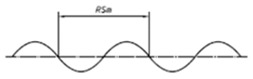
Type C2	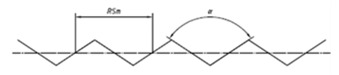
Type C3	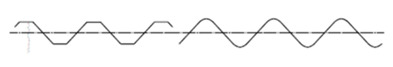
Type C4	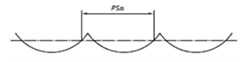
Type D1	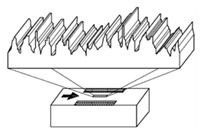
Type D2	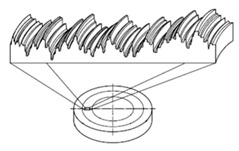
Type E2	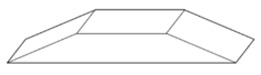

**Table 3 materials-14-00029-t003:** Areal surface texture artifacts according to ISO 25178-701 [1,40].

**Standard**	**Scheme**
Type ER1	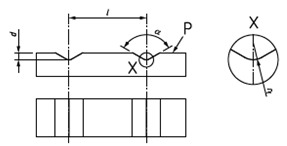
Type ER2	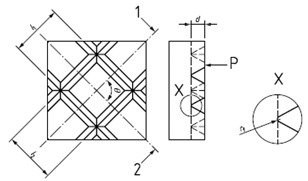
Type ER3	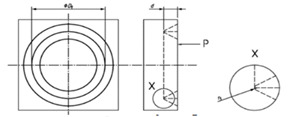
Type ES	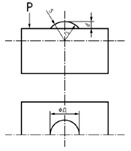
Type CS	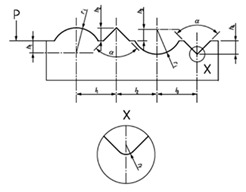
Type CG1	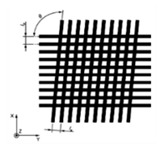
Type CG2	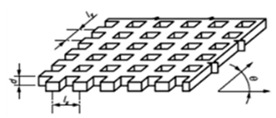

## Data Availability

The data presented in this study are available on request from the corresponding author. The data are not publicly available due to they are part of an ongoing, wider research.

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
