# Peer review of "Design of Industrial Standards for the Calibration of Optical Microscopes"

_materials, 2020, doi:10.3390/ma14010029_

Round 1

Reviewer 1 Report

The article contains a comprehensive description of the available roughness reference workpieces.

This work would be of much greater value if it contained measurement results from the measurement of the standards proposed in the article, using various measuring devices.

The description of the standards is not enough, but it can be treated as a starting point for further work.

Additionally, the suggestion of using standards in destructive testing requires additional comment (line 577). Probably the authors do not want to apply the grooves of standards on samples for tensile tests, which would weaken the samples and lead to damage in the places where the grooves occurs.

Author Response

Dear Reviewer

First of all, we would like to thank you for your work as a reviewer and for the comments and the feedback you have given us. They have allowed us to clearly improve our manuscript.

We would also like to thank you for your positive view of the paper regarding the description made about the different reference material standards.

Following your suggestion:

  • We have included some measurement results obtained over two types of standards, one commercial stage micrometer and a custom bidimensional standard manufactured at UPM Laser Center (new sections 4.2 and 4.3). But because of lack of time, we have not been able to include comparative measurements of the same standard in different instruments. In any case, our first objective was only to demonstrate the feasibility of manufacturing custom material standards with free form geometries as it can be done, for example, with direct laser writing.
  • We have included an improved description of some standards (pages 11 and 12)
  • We have clarified that, in destructive tests, these standards can be used after the test has been done, to perform dimensional measurements.

Sincerely yours,

Alberto Mínguez

Jesús de Vicente

Reviewer 2 Report

The paper raises the problem of traceability for surface topography measurement. Authors present the proposition of designs of material standard that can be used  for optical measuring instruments calibration.

The first part of the work, which contains an overview of the methods, the discussion of standards, is generally interesting. However, each one of the cited references within the body of the paper should be discussed individually and explicitly to demonstrate their significance to the study. In some cases, it is of course not possible (e.g. referring to several standards or catalog notes), but the authors of the cited scientific or book publications should be discussed individually.

As for the readability of the work, I would suggest to consider replacing Fig. 2 with a table containing drawings of surface textures arranged in rows according to a given type.

While the discussion of the NPL standard was sufficient, for the methods in lines 298 – 342 apart from the photos there is practically no description - it is definitely not enough.

Line 384: citation – [27-40] would be better.

Generally, work up to section 2.2. does not raise any major objections apart from those mentioned. However, the rest, intended to be an essential part of the work, seems to be underdeveloped.

section 2.2 is confusing - what do the lines in Fig.10 mean and why is this drawing introduced with the description "comercial stage micrometer (...) dismensions:"?

The concept presented by the authors is interesting, however, at the present stage, only the geometrical properties of the models are presented, which is not yet sufficient for publication. I believe that the article will be fully valuable only when the selection of materials and the technology of making models are also presented.

Author Response

Dear Reviewer

First of all, we would like to thank you for your work as a reviewer and for the comments and the feedback you have given us. They have allowed us to clearly improve our manuscript.

We would also like to thank you for your positive view of the paper regarding the overview of the methods and the discussion of the standards.

Following your suggestions:

  • We have included a paragraph about MATWEB and other references to books, papers, manufacturers brochures, etc… have been briefly commented on.
  • We have replaced figure 2, as you suggested, with a new table on page 7. We think the information now is clearer.
  • We have included more information about standards from PTB and SiMetrics on pages 11 and 12.
  • The citation in line 384 has been shortened.
  • We have included more information about the models on pages 12 to 16.
  • We have reworded the caption of figure 10 and the references to stage micrometers.
  • We have included more information about material selection y manufacturing processes in section 4.
  • As a proof of concept (section 4.3), we have included results from a custom standard manufactured at UPM Laser Center in comparison with the groove geometry of a commercial metallic stage micrometer.

Sincerely yours,

Alberto Mínguez

Jesús de Vicente

Reviewer 3 Report

The manuscript has very little novelty. An improved version of the manuscript might be more suitable for a Metrology journal. In the current form, the manuscript has no benefit to the readers of this journal. The manuscript needs a better structure (e.g. "Materials and Method" might not be necessary/or could be renamed). The aim of the study is not clearly stated at the beginning.

Author Response

Dear Reviewer

First of all, we would like to thank you for your work as a reviewer and for the comments and the feedback you have given us. They have allowed us to clearly improve our manuscript.

We know that this work could be classified as a classical topic of dimensional metrology. But we think that surface texture is becoming more and more important when talking about material properties. And, in the special issue of SIMES-2020 at Materials journal, there was a topic included in it called “metrology and quality in manufacturing”. That was the reason why we finally decided to send this work to this Special Issue.

Following your suggestions:

  • We have tried to clarify the objectives of the work (page 2)
  • We have improved the structure of the paper. Some paragraphs have been relocated and we have improved the wording of the text to make it clearer.
  • As a proof of concept (section 4.3), we have included results from a custom standard manufactured at UPM Laser Center in comparison with the groove geometry of a commercial metallic stage micrometer.

Sincerely yours,

Alberto Mínguez

Jesús de Vicente

Round 2

Reviewer 2 Report

The authors satisfactorily responded to comments in the previous review and significantly improved the article. Although the study is still not fully exhaustive, it meets the requirements for scientific articles in its current form.

Reviewer 3 Report

The manuscript has been improved and can be publish (English should be brushed up a bit).